

# Contradiction in text review and apps rating: prediction using textual features and transfer learning

Turki Aljrees[1], Muhammad Umer[2], Oumaima Saidani[3], Latifah Almuqren[3], Abid Ishaq[2], Shtwai Alsubai[4], Ala' Abdulmajid Eshmawi[5] and Imran Ashraf[6]

[1] College of Computer Science and Engineering, University of Hafr Al-Batin, Hafar Al-Batin, Saudi Arabia

[2] Department of Computer Science, Islamia University of Bahawalpur, Bahawalpur, Punjab, Pakistan

[3] Department of Information Systems, College of Computer and Information Sciences, Princess Nourah bint Abdulrahman University, Riyadh, Saudi Arabia

[4] Department of Computer Science, College of Computer Engineering and Sciences, Prince Sattam bin Abdulaziz University, Al-Kharj, Saudi Arabia

[5] Department of Cybersecurity, College of Computer Science and Engineering, University of Jeddah, Jeddah, Saudi Arabia

[6] Department of Information and Communication Engineering, Yeungnam University, Gyeongsan, Republic of Korea

Corresponding authors
Muhammad Umer,
umersabir1996@gmail.com
Imran Ashraf, imranashraf@ynu.ac.kr

## ABSTRACT

Mobile app stores, such as Google Play, have become famous platforms for practically all types of software and services for mobile phone users. Users may browse and download apps *via* app stores, which also help developers monitor their apps by allowing users to rate and review them. App reviews may contain the user's experience, bug details, requests for additional features, or a textual rating of the app. These ratings can be frequently biased due to inadequate votes. However, there are significant discrepancies between the numerical ratings and the user reviews. This study uses a transfer learning approach to predict the numerical ratings of Google apps. It benefits from user-provided numeric ratings of apps as the training data and provides authentic ratings of mobile apps by analyzing users' reviews. A transfer learning-based model ELMo is proposed for this purpose which is based on the word vector feature representation technique. The performance of the proposed model is compared with three other transfer learning and five machine learning models. The dataset is scrapped from the Google Play store which extracts the data from 14 different categories of apps. First, biased and unbiased user rating is segregated using TextBlob analysis to formulate the ground truth, and then classifiers prediction accuracy is evaluated. Results demonstrate that the ELMo classifier has a high potential to predict authentic numeric ratings with user actual reviews.

## INTRODUCTION

The prevalence of cell phones has impacted professional and domestic lives. Millions of apps related to beauty, medical care, surveillance, fitness, education, entertainment, and so

on are available to aid in different aspects of daily life. Figure 1A shows that there are over 2.67 million apps accessible on the Play Store, a platform for Android OS from Google, (*Statista, 2023a*) as of March 2023. These apps are downloaded, utilized, and rated by the users based on their experiences and functionalities offered by these apps in numerical terms as well as feedback reviews. As indicated in Fig. 1, over 612.2 billion apps are being downloaded annually, and the figure is predicted to rise to 700 billion by the end of 2025 (*Statistaa, 2023b*).

Opinion mining is a technique to locate and analyze personal information by using computational linguistics, text analysis, and natural language processing (*Zong, Xia & Zhang, 2021*). Some popular opinion-mining techniques are lexicon-based approaches, Aspect-Based Sentiment Analysis, and rule-based approaches used in sentiment analysis, language translation and text summarization (*Pimpalkar & Jeberson Retna Raj, 2021*). It is a subdomain of text mining utilized to recognize and retrieve the desired information from text by creating automated systems. The decision-making process is influenced greatly by opinions. User ratings and reviews act as a beacon house for other users when deciding to download or utilize these programs. According to research (*Horrigan, 2008*), user evaluations and numerical ratings have a significant impact on mobile app adoption in general. Users prefer to buy a high-rated app over another app *i.e.,* a 5-star app over a 4-star rate app, even if there is a pricing difference of 25%–95%. Users' and app developers' participation broadens with reviews, ratings, and issue reports (*Li et al., 2023*).

As seen in Fig. 2, user input is presented in two formats: text reviews and numerical ratings. Numerical ratings are awarded as stars while a text review may include negative or positive comments on a certain app or policy by the user. Public relations management, net promoting scoring, marketing analysis, product feedback, product reviews, and customer services focus on benefitting from this data (*Sundararaj & Rejeesh, 2021*).

A numerical number is assigned by the user ranging from 1 to 5 in a numeric rating usually defined and summed as a star rating for an app on an Android Play Store. A higher rating attracts more users to select an app and install it. The prospect of fraudulent, biased, and fake reviews poses a big challenge. These biased or fraudulent reviews can lure users to download such apps based on their star ratings (*Dou et al., 2019*). Currently, no standard procedure is in place to verify the numerical ratings legitimacy. This results in annoyance and ambiguity for mobile app users looking for top-notch mobile apps. Most users go through a few recent comments on an app before they decide to download or purchase the app. Typically, time-pressed consumers do not have the luxury of reading whole evaluations, which leads them to select the incorrect app.

Although numerous ways to deal with these issues have been offered (*Thiviya et al., 2019*; *Hu et al., 2019*), all these methods evaluate user reviews to find just a bias inherent in user reviews. Subjectivity and polarity are characterized by *neutral*, *positive*, and *negative* descriptors. These approaches can lead to misaligned numerical ratings (*Shashank & Naidu, 2020*) because of no consideration for actual app ratings and their sole focus is on user reviews. We found that contradicting opinions can be the outcome of non-synchronized user assessments and numerical ratings awarded to certain apps.

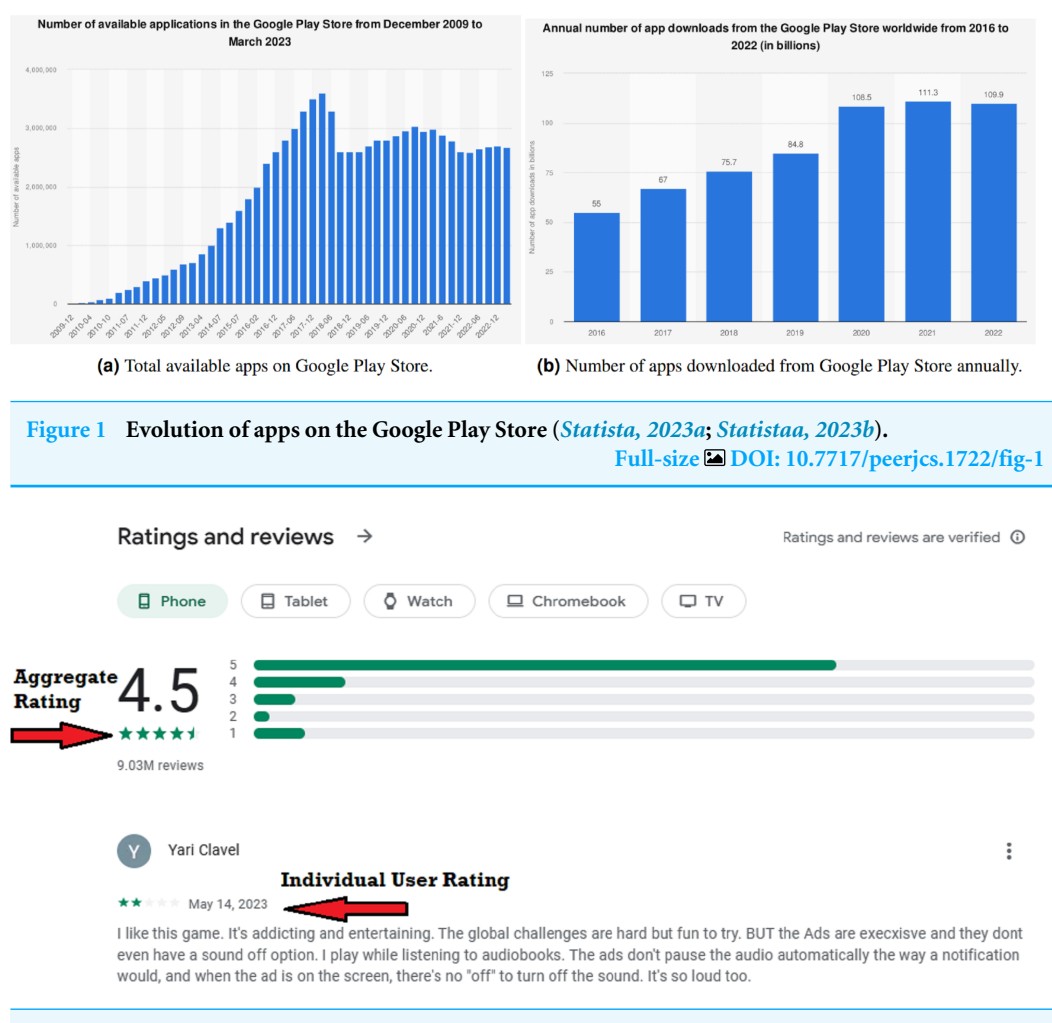

**(a)** Total available apps on Google Play Store.

**(b)** Number of apps downloaded from Google Play Store annually.

**Figure 1** Evolution of apps on the Google Play Store (*Statista, 2023a*; *Statistaa, 2023b*).

**Figure 2** Example of aggregate and individual user rating on Google Play Store.

This research proposes an approach to circumvent this limitation by predicting a mobile app's numerical rating based on user comments. This is performed by using an ensemble learning (*Zhou et al., 2021*) and transfer learning classifier (*Zhuang et al., 2020*). Preprocessing of the dataset includes *case transformation*, *removing numbers*, *removing stop words*, *stemming*, and *tokenization* (*HaCohen-Kerner, Miller & Yigal, 2020*). We gathered about 25,000 archives related to each target rating to balance the dataset. It helps to overcome the issue of training a model on disproportional data containing a higher number of 5-star rated examples. 125,000 instances have been reported as per the new dataset. Two types of feature representation have been used in this research work hand-crafted term frequency/inverse document frequency (TF/IDF) (*Jalilifard et al., 2021*) and pre-trained word vector representation bidirectional encoder representations from Transformers (BERT) (*Jwa et al., 2019*), robustly optimized BERT (RoBERTa) (*de Oliveira, 2022*), XLNet (*Rajapaksha, Farahbakhsh & Crespi, 2021*), and embeddings from language model (ELMO) (*Rehman, Sanyal & Chattopadhyay, 2023*). Vector-space modeling (VSM)

technique of TF/IDF is applied to the preprocessed data, taking into account unigrams, bigrams, trigrams, and TF. These algorithms produce features that are then fed into learning models. The following are the important contributions of this research

- This research work is divided into two phases. In the first phase, this study analyzes the classification results of machine learning and transfer learning models. In the second phase, the best-performing model is used to analyze the discrepancy between actual user reviews and numeric ratings.
- To predict numeric ratings, machine learning techniques such as the gradient boosting machine (GBM), random forest (RF), AdaBoost classifier (AB), the extreme gradient boosting (XGB) classifier, and the extra tree classifier (ET) are used.
- To predict numeric ratings, transfer learning techniques such as BERT, robustly optimized BERT (RoBERTa), XLNet, and ELMO are used. Classifiers are examined with the review text features to predict the actual numeric rating.
- Prominent apps are validated from each category for comparing numeric ratings predicted by our model with the actual ones from the users.

The research is structured into five sections. The literature has been reviewed in the following section. It is followed by the description of the ensemble learning classifiers utilized in this study. Afterward, the results are discussed. In the end, the study is concluded with future research directions.

## RELATED WORK

The amount of personal and public data on the internet has been gradually increasing over the past decade. People put textual information on blogs, forums, review sites, and other social media platforms. This unstructured input can be automatically converted into structured data that reflects public opinion using review-based prediction systems. This data can portray user sentiments towards particular brands, applications, goods, and services (*Aslam & Ashraf, 2014*; *Rashid et al., 2013*). As a result, they can offer crucial information for improving products and services. The subsequent investigations used this form of sentiment analysis.

Sentiment analysis has versatile applications in market intelligence. It can gauge user satisfaction levels regarding products or services, pinpoint areas for improvement, predict price fluctuations based on sentiment in news, facilitate the creation of new products and services, and enhance marketing strategies based on customer reviews (*Tubishat, Idris & Abushariah, 2018*). The process involves collecting reviews, detecting sentiments, categorizing them, selecting relevant features, and determining sentiment polarity. These reviews are subjected to various technical processing and classification techniques for sentiment analysis (*Rahul, Raj & Monika, 2019*). These reviews and ratings encompass descriptions that span both positive and negative aspects (*Dharaiya et al., 2020*). Hence, it is crucial to discern the sentiment of customers' opinions in social media, whether they are positive, negative, or neutral (*Meena, Mohbey & Indian, 2022*).

The identification and organization of opinions in the text have also been examined using information-extraction tools. For example, opinions are represented by annotating the text

in *Yimam et al. (2020)*. The authors also described a "scenario template" to summarize documented opinions that are opinion-oriented. This method is useful for activities that require asking questions from several angles. The semantic orientations of words could be extracted using a model based upon statistical analysis, according to *Misuraca, Scepi & Spano (2021)*. The approximation probability in the spin model is calculated using mean-field approximations. The suggested model generates extremely precise semantic directions based on the English lexicon from a reduced set of seed words.

In addition to the above-mentioned approaches, various techniques are used for sentiment analysis (*Yadav & Vishwakarma, 2020*). These issues are addressed with sentiment analysis which employs mathematical and statistical techniques, particularly those utilizing Gaussian distributions. A machine learning (ML) algorithm is used to predict a Google app ranking using a dataset that includes the number of downloads and feedback reviews, app category, type of the app, version of an app, and the app size as well as the content rating. Support vector machine (SVM), artificial neural networks (ANN), k-means clustering, linear regression, decision trees, logistic regression, NB classifier, and k-nearest neighbours are investigated in this regard.

Based on different feature extraction approaches, app ratings have been predicted (*Suleman, Malik & Hussain, 2019*; *Sarro et al., 2018*). Experiments are performed for BlackBerry World and the Android app store using basic attributes such as cost, textual descriptions, ratings, and popularity among downloaders. These features are numerically vectorized to predict app ratings and to support case-based reasoning. The views posted on Google app reviews are studied by other authors (*Martens & Johann, 2017*) in contrast to the aforementioned research. Their research evaluated the ideas and feelings expressed in user reviews using a range of emojis that could signify, for instance, rage, enthusiasm, positivity, or negativity. It assessed whether such opinions are instructive for the improvement and development of the app. The process of creating and annotating emotional dictionaries traditionally relies on manual annotation, a labour-intensive and time-consuming task, particularly when multiple labels are needed for each word. Automated emotion annotation remains relatively uncommon. In *Mehra (2023)*, a machine learning-driven iterative design approach is introduced to automate the prediction of user satisfaction within the Smart product service system. In another work, the authors classify hate speech using an attention-based deep learning model (*Fazil et al., 2023*).

Despite the results and efficacy reported in these studies, these approaches are inadequate in several ways and are inappropriate for predicting numerical ratings of Play Store apps. Text-mining techniques are not appropriate to evaluate Play Store apps due to the restricted quantity of words and Unicode-supported language. Such research utilizes rating projections based on app characteristics or features *e.g.*, bug reports, price, *etc.* No research looked into potential conflicts between user reviews and numerical ratings. This is the first study to look at these differences for numerical-rating predictions for apps on the Play Store for Android devices based on user reviews.

# MODELS USED FOR PREDICTING NUMERIC RATING

This research work uses machine learning classifiers to predict the numerical rating of mobile apps based on user reviews. The performance of several machine classifier learning models is assessed in this study, which are briefly described below. *Scikit learn* (*Hackeling, 2017*; *Scikit Learn, 2023a*) is used in Python to implement the models. Since machine learning approaches incorporate various ML methods into one prognostic model, they assist in reducing bias (boosting), and variance (bagging), and enhancing predictions (stacking) (*Araque et al., 2017*).

## Machine learning models

RF creates numerous decision trees, which lowers the variance, but the results are difficult to understand. Regression and classification are two uses of RF. Results are combined into a single final result using a group of decision trees. By using a subset of characteristics randomly and training them on various data samples, variance is reduced by RF in a couple of different ways (*Breiman, 2001*).

AdaBoost is the very first algorithm that outshines other related algorithms in boosting classification tasks concerning accuracy. AdaBoost is an adaptive model as it takes into account prior misclassifications made to adjust for weak learners (*Schapire & Singer, 1999*). AdaBoost attains a high level of accuracy when taking into account a fully learned model. Computing a feature significance score is also utilized to identify key characteristics.

For classification and regression issues, GBM is applied. It utilizes several weak learners like decision trees to create a prediction model (*Natekin & Knoll, 2013*). Boosting is a technique in which weaker models are turned into stronger ones. Boosting is characterized by a tree model in which every tree is adjusted to the original dataset in an updated/modified state. Loss functions use gradients to evaluate the performance of coefficient models to give GB its strong performance. Depending on the amount, feature, or quality that has to be optimized, the loss function's exact specification will change.

Due to its efficiency and quick execution, the boosting method XGBoost is quite popular among data scientists. Its implementation, which consists of stochastic gradient, regularized gradient boosting, gradient boosting, and boosting is what gives it its high performance. Both classification and regression issues are addressed by it. As XGBoost helped multiple Kaggle tournament winners succeed, it received higher attention (*Chen & Guestrin, 2016*). As in other bagging and boosting classifiers, feature significance scores are also computed.

To increase prediction accuracy and reduce overfitting, the ET classifier incorporates a meta-estimator that fits randomized decision trees (RT) on different sub-samples of a dataset (*Geurts, Ernst & Wehenkel, 2006*). The ET classifier uses the complete sample at each step in contrast to RF and chooses decision limits at random rather than choosing the best one. The phrase "very randomized tree" is another name for it.

## Transfer learning models used for predicting numeric rating

This research work also analyzes the performance of state-of-the-art transfer learning models.

### Bidirectional encoder representation from transformers

BERT revolutionized natural language processing (NLP) with its innovative implementation of a highly effective bidirectional self-attention mechanism. This mechanism is trained on an extensive collection of data, including the BookCorpus, which encompasses 11,038 unpublished books in plain text format across 16 diverse genres. Additionally, it utilizes 2,500 million words extracted from English Wikipedia passages to further enhance its capabilities (*Rajapaksha, Farahbakhsh & Crespi, 2021*). Unlike context-free models such as Word2Vec, BERT utilizes a bidirectional contextual model that takes into account both the preceding and following words in a sentence. Consequently, contextual models capture different word representations based on the specific sentence context, offering a more comprehensive understanding of language nuances. In contrast, context-free models assign the same representation to a given word regardless of its context in different sentences. The BERT model undergoes training on diverse unlabeled data corpora, encompassing various scenarios. In the subsequent fine-tuning phase, the model starts with pre-trained parameters as its initialization. BERT utilizes the "[MASK]" symbol to predict missing tokens in the text. However, BERT does have a few noteworthy drawbacks. Firstly, the reconstruction of all masked tokens and corrupted versions in the joint conditional probability is conducted independently, which can be seen as a limitation. Secondly, masked tokens do not appear in the downstream tasks, resulting in a disparity between the pre-training and fine-tuning stages. However, one of the significant advantages of BERT's autoencoder (AE) language modeling approach is its ability to capture bidirectional context, enabling a more comprehensive understanding of language.

### Robustly optimized BERT approach

RoBERTa, an enhanced version of BERT, shares many similar configurations with BERT but shows improved performance (*Rajapaksha, Farahbakhsh & Crespi, 2021*). This can be observed from the GLUE leaderboard. RoBERTa surpasses BERT in performance by implementing several significant modifications. These changes involve leveraging a larger training dataset, adopting dynamic masking patterns, training on lengthier sequences, and replacing the next sentence prediction task. In essence, RoBERTa fine-tunes BERT by primarily augmenting the data size and optimizing hyperparameters. In RoBERTa, dynamic masking is applied to every training instance during each epoch. This is accomplished by replicating the training dataset ten times, resulting in each sequence being masked in ten distinct ways throughout forty training epochs.

### XLNet

XLNet is a transfer learning model developed by Google AI in 2019. It is similar to BERT but utilizes the AutoRegressive (AR) pre-training method, resulting in improved performance compared to BERT on various benchmark datasets (*Rajapaksha, Farahbakhsh & Crespi, 2021*). XLNet addresses autoencoder (AE) model limitations through bidirectional context, permutation-based training, larger size and data, pretraining and fine-tuning, attention mechanisms, and strong performance across diverse natural language processing tasks. These features enable XLNet to capture richer contextual information, long-range dependencies, and complex linguistic patterns, making it more effective for understanding

and generating natural language text (*Su et al., 2021*). It introduces permutation language modeling (PLM) as a solution. In XLNet, permutations of occurrences for a given word are used, allowing the model to train on every possible word sequence. However, this approach leads to a longer convergence time compared to BERT due to the increased complexity of considering all permutations. The primary concept behind XLNet is to utilize PLM to enhance the capturing of bidirectional contexts. In XLNet, if a sentence consists of $x$ tokens with a length of $T$, a total of $T$ different orders can be generated by considering all positions on both sides of a token for AR factorization. Let $Z_T$ be the set of all possible permutations of sequences with a length of $T$, then

$$maxE_Z \sim z_T[\sum_{t=1}^{T} logp(x_{zt}|X_{z<t})]. \tag{1}$$

In XLNet, the permutation is applied to the factorization order rather than the sequence order. The original sequence order is retained, and Transformers are used to incorporate positional encoding that corresponds to the original sequence. This characteristic of XLNet, where the original sequence order is preserved during pre-training, proves to be beneficial for fine-tuning tasks, as it allows the model to focus on the natural order of the given sequence.

### Embeddings from language models

Traditional word embedding methods often struggle to capture contextual information and accurately distinguish between polysemous words (*Huang & Zhao, 2020*). As a result, these methods tend to generate the same representations for words like "read" regardless of the specific context in which they appear. In contrast, word embeddings derived from ELMo align with the contextual nuances of different sentences. These embeddings are generated by leveraging the learned functions of all the internal layers within a bidirectional Long Short-Term Memory (LSTM) model. As a result, the representations of the word "read" in different contexts vary, capturing their unique contextual usage. ELMo provides notable benefits in generating contextualized representations, making it essential to utilize ELMo embeddings for various NLP tasks, particularly those involving text similarity calculations.

# PROPOSED APPROACH FOR APPLICATIONS RATING EVALUATION

The suggested strategy, components, as well as test dataset are all described in this section. Figure 3 depicts the design of the suggested method for predicting numeric ratings while Fig. 4 shows the complete workflow methodology diagram of the proposed approach. It entails several sub-modules, each of which is detailed below.

## Dataset used for experiments

Using the web scraper *BeautifulSoup*, the Google applications dataset is extracted from the Google Android Play store. The data is collected for the last five years. The following standards are used

1. The app must be at least five years old.

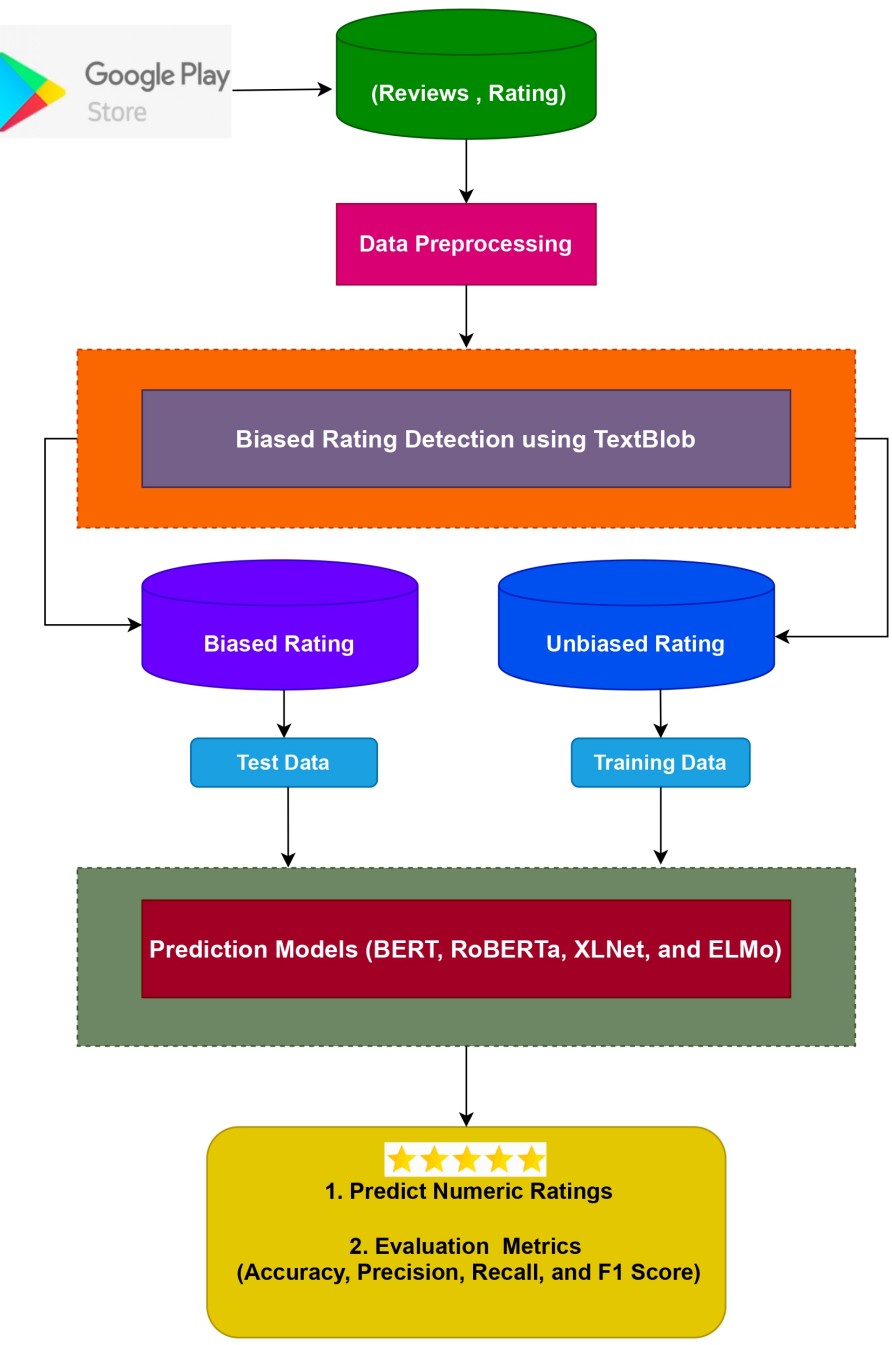

**Figure 3** Architecture of the proposed approach.

2. There must be at least 4,000 reviews on the app.

The dataset comprises several features including 'app id', 'app category', 'app rating', 'app name', and 'app review'. Table 1 provides names and descriptions of these features. The total count of extracted data is 502,658. This research focused on examining the inconsistency between reviews and rating estimates centered upon the attributes collected

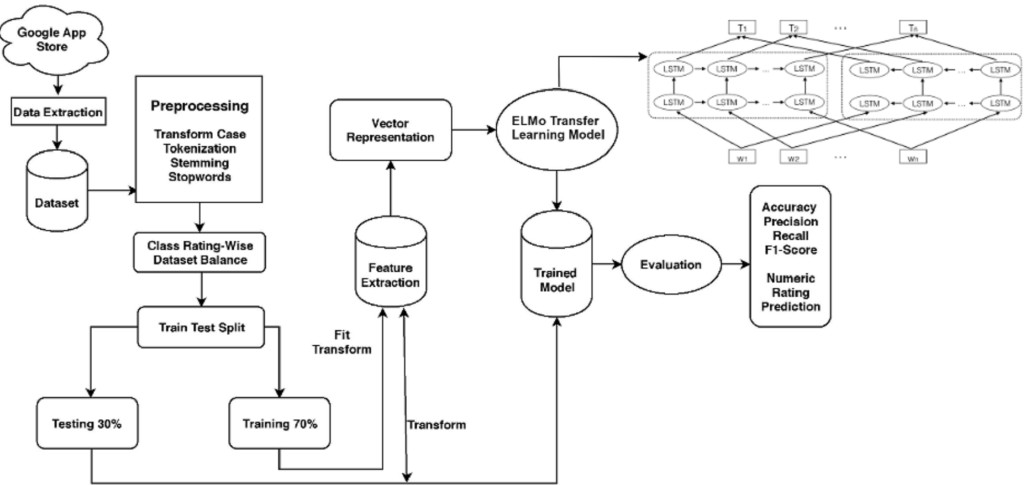

**Figure 4   Workflow methodology of the proposed approach.**

**Table 1   Dataset features and description.**

| Features | Description |
|---|---|
| Application category | It represents the category of the app in the Google Play store. |
| Application name | It shows the actual app name. |
| Application id | It represents the unique app id; this id is assigned to the app for quick identification. |
| Application review | It represents the user's review about the specific app. |
| Application numeric rating | It shows the rating given by the app users on the Google Play Store. |

from the reviews. Other variables and associated meta-data are also available on the Play Store. However, we ignored other features and meta-data that are unnecessary.

There are 14 different types of mobile applications in the scraped data. To cover the wide range of applications, sampling across many categories is done primarily for that reason. The numerous forms of textual evaluations in each area such as sports, news, entertainment, health, and business use a wide range of idioms and terms. So, rather than focusing on a small number of categories, the aim is to precisely estimate the classification accuracy by testing the classifiers performance and applying them to a variety of reviews. A few sample reviews from sports, communication, weather, action, and health & fitness categories are displayed in Table 2.

All of the categories in the dataset are listed in Fig. 5 along with their relative sizes. Each app has almost 4,000 reviews in the scraped database. Individual user data on app reviews and ratings are examined.

The flow chart in Fig. 6 shows how the data is collected employing the *BeautifulSoup* (BS) web scraper. By seeing trends, patterns, and connections, it is found that they could go unnoticed in data based on text, data visualization is crucial for comprehending the

**Table 2  Example category reviews used in the dataset.**

| App category | Reviews |
| --- | --- |
| Sports | I just wanted a basic billiards game. There is this redundant tutorial level-up system that is totally unnecessary and a map thing to show progress and weird pool tables. Political ads galore. Just stop. |
| Communication | I try this app I feel like a real messenger |
| Action | I like the graphics it is a good choice |
| Weather | Useful, but I would like to see the weather patterns over South Africa showing an approaching cold front egg. |
| Health & Fitness | The only way I can do any workouts)) Obviously, it's not ideal, especially the Russian version. Also Hungarian one would be awesome, but in general - LOVE IT? |

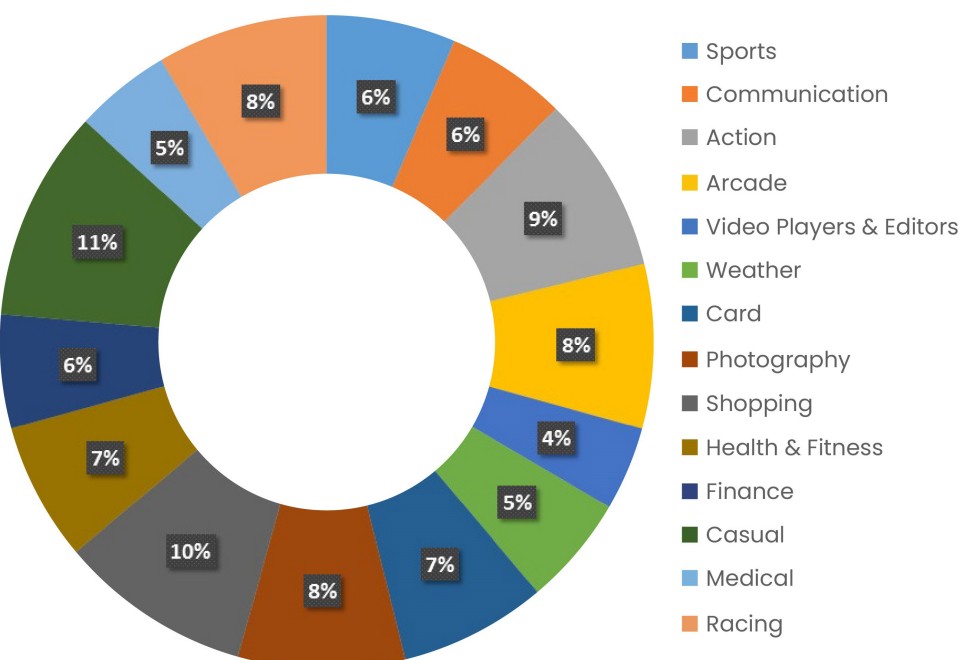

**Figure 5  Category-wise number of reviews in the dataset.**

dataset. Data visualization makes it simple to spot these linkages in various datasets. To present the numerical ratings given to mobile apps, we displayed the dataset.

Figure 7 shows that five stars are the most common app user rating. Yet, a significant worry is the potential for slanted or even false ratings provided by anonymous people. So, we took into account how frequently each category received numerical ratings.

Figure 8 demonstrates that casual mobile apps consistently receive better scores, ratings, and/or rankings and are expected to exhibit a 5-star rating from the users of the app. The lowest numerical ratings, on the other hand, are typically given to creators of video players and editors.

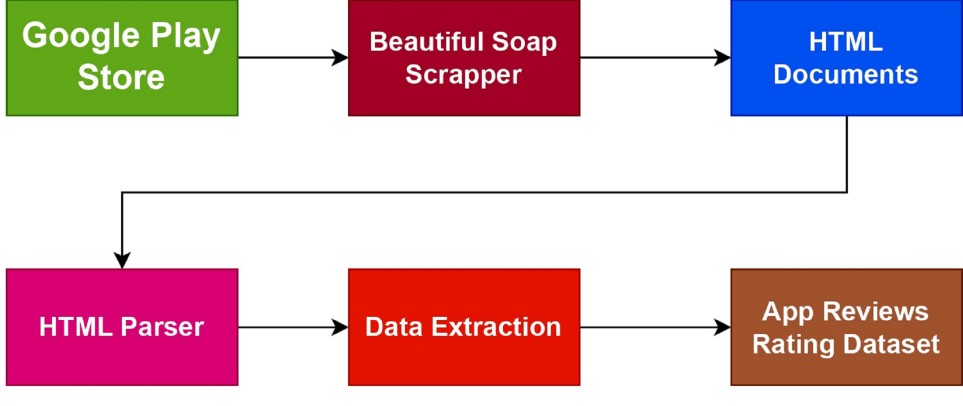

**Figure 6   Framework of data scrapping.**

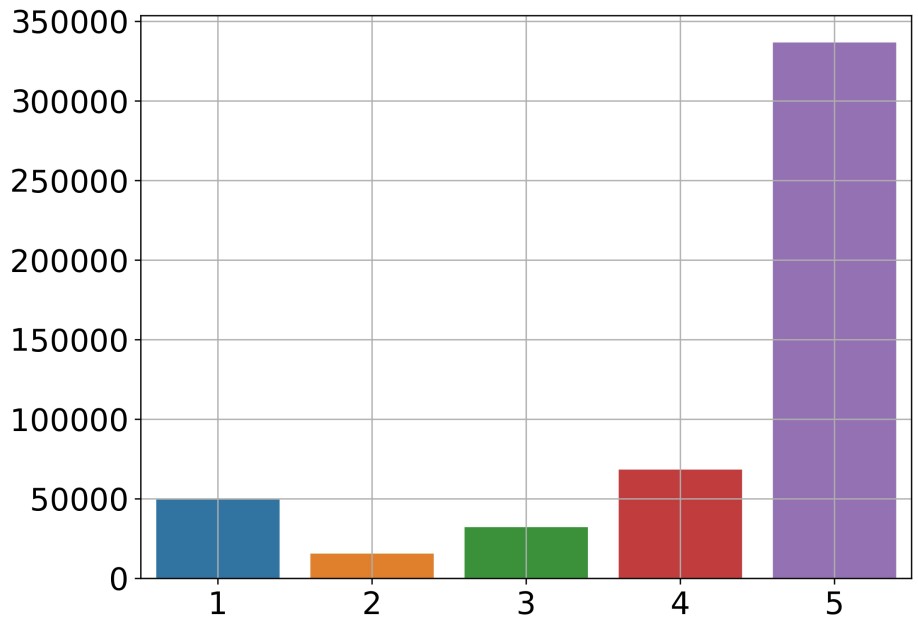

**Figure 7   Most frequent ratings given from users.**

## Reviews pre-processing

The dataset related to the Android Play Store is either unstructured or semi-structured and hence holds a lot of unnecessary information that does not significantly contribute to the prediction. Text pre-processing is a prerequisite to avoid this limitation since extended training is needed for large datasets while 'stop words' lower predicted accuracy. Pre-processing entails many processes such as stemming, lowercase conversion, punctuation, and removing terms that do not have a greater significance concerning the text. According to research, text preparation significantly increases prediction accuracy (*Feldman & Sanger, 2007*). So, before beginning the training process using the suggested technique, the data

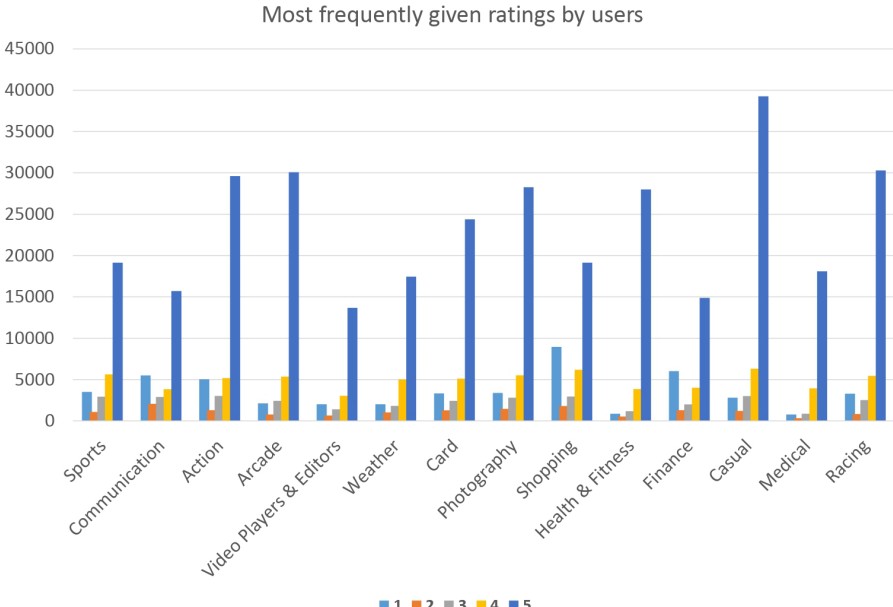

**Figure 8**  **Each category ratings from users.**

**Table 3**  **Working example of reviews pre-processing steps used in the dataset.**

| Pre-processing | Output |
| --- | --- |
| Actual text | 'an\u003e' Awesome, love the pictures and description of each medication. ' |
| Transform to lower case | 'an\u003e' awesome, love the pictures and description of each medication. ' |
| Remove HTML tags | awesome, love the pictures and description of each medication. ' |
| Remove stop words & tokenization | [awesome, love, pictures, description, each, medication, . '] |
| Stemming | awesome love picture description each medication. |

must be normalized. The use of the smartphone app, *Seven 7 min workout challenge*, is reviewed as an illustration for exercise challenge as a text pre-processing to have a better understanding of it. Table 3 displays a series of pre-processing steps and their results. The ensemble learning classifiers may be used on the processed dataset when pre-processing is finished.

## Feature engineering

To train supervised machine learning algorithms, textual materials need to undergo vectorization. This process involves converting text into numerical representations while preserving all the relevant information. Various methods can be employed for text transformation, such as the Bag-Of-Words (BOW) approach, in which a unique number is assigned to each word. However, BoWs effectiveness is compromised due to limitations on character length in reviews. Additionally, the accuracy of BOW-based

**Table 4  Working example of feature generation using TF technique.**

| Fantastic | Game | Kid | Appreciate |
|---|---|---|---|
| 1 | 1 | 1 | 1 |

**Table 5  Working example of feature generation using TF/IDF technique.**

| Fantastic | Game | Kid | Appreciate |
|---|---|---|---|
| 0.31 | 0.31 | 0.31 | 0.31 |

techniques is hindered by inadequate occurrences of certain words (*Sriram et al., 2010*). To address these challenges and accomplish the transformation, we utilize the concept of term frequency (TF) (*Scikit Learn, 2023b*). This approach considers the number of occurrences of a word(s) in a certain document and generates a matrix that represents the overall frequency of each word across the entire document. To illustrate the application of TF, let us consider the *My Talking Tom* app. The initial text is:

*It's fantastic, and my kids adore playing it. I appreciate that you have this game in the shop.*

Preprocessing turns the review into a *fantastic game kid appreciate. I appreciate the gaming shop*. In Table 4, the matrix produced by using TF is displayed.

In the proposed technique, feature selection is done using the TF/IDF (uni-, bi-, and trigrams) (*Scikit Learn, 2023c*). TF-IDF assigns higher weights to terms that appear frequently in a text subset while giving lower weights to words that are extremely frequent and appear in almost all documents. So, terms that are used frequently are given a lesser weight than uncommon words that are used more specifically in a given document. Table 5 displays the TF/IDF result for the aforementioned scenario.

## Accuracy measures

To evaluate performance, we employed many metrics. Understanding the various accuracy characteristics of a classifier requires a comprehension of four fundamental concepts (*Han, Pei & Kamber, 2011*).

- True positives (TP) refer to the instances where the positive class is accurately predicted by the classifier.
- True negatives (TN) are instances where a negative class is correctly predicted by the classifier.
- False positives (FP) occur when the classifier incorrectly labels negative instances as positive.
- False negatives (FN) are the instances mistakenly marked as negative by the classifiers which are positive instances.

In many classification methods, accuracy is a key assessment factor. Accuracy is a measure of how many correctly predicted instances (both true positives and true negatives) there are among all the instances in the dataset. It provides an overall assessment of the model's correctness. In terms of the previously defined TP, FP, TN, and FN, it is computed

as follows

$$accuracy = \frac{TP + TN}{TP + TN + FP + FN} \times 100 \qquad (2)$$

Accuracy measures overall correctness in predictions, while precision specifically quantifies the model's ability to avoid false positive predictions.

$$precision = \frac{TP}{TP + FP} \qquad (3)$$

Recall, also known as sensitivity or true positive rate, measures the ratio of true positive predictions to all actual positive instances in the dataset. It tells us how many of the actual positive instances were correctly predicted by the model. Recall is calculated as:

$$recall = \frac{TP}{TP + FN} \qquad (4)$$

The accuracy and recall are taken into account while calculating the F score, which has a range of 0 to 1. It considers both precision and recall and is regarded as a more important metric, especially when the dataset is imbalanced. It is computed using

$$F_1 = 2 \times \frac{precision \times recall}{precision + recall} \qquad (5)$$

## EXPERIMENT AND RESULTS

This section describes the experiments carried out to evaluate the performance of all models. The experiment involved creating a dataset consisting of reviews from 168 applications. To form this dataset, 14 app categories are selected, with each category containing 12 different apps. The experiment is conducted to predict the numeric rankings of these apps and use ensemble learning models. The Google Play store ratings used to determine the app categories' popularity are taken into consideration. In comparison, one mobile application from each category is chosen based on the number of reviews and overall rating. The experiment's categories and the applications chosen for each category are listed in Table 6.

### Performance evaluation of machine learning models

Numerous experiments are conducted to assess the effectiveness of the selected machine learning classifiers, as well as the transfer learning classifiers. The experiments are divided into three groups: experiments involving TF features, experiments involving TF/IDF features, and experiments involving transfer learning models.

Table 7 displays the performance of machine learning models using TF features. The results demonstrate that the ET model outperforms other models significantly achieving a 75% accuracy score due to its ensemble-boosting architecture. Even with small amounts of data, the ET model excels in accuracy compared to all other models. XGB follows closely behind with a 74% accuracy score, indicating that linear models can also perform well with TF features. GBM, on the other hand, exhibits the lowest performance, achieving a 70% accuracy score.

**Table 6  Google apps data selected for the experiment.**

| Application category | Selected application | Total reviews |
|---|---|---|
| Arcade | Tempe Run 2 | 40,751 |
| Action | Gun Shot | 44,141 |
| Card | Teen Pati Gold | 36,520 |
| Communication | UC-Browser | 30,000 |
| Casual | Candy Crush Saga | 52,560 |
| Finance | PhonePe | 28,220 |
| Health & fitness | Seven | 34,415 |
| Medical | Pharmapedia Pakistan | 24,002 |
| Photography | B612 | 41,440 |
| Racing | Beach Buggy Racing | 42,384 |
| Shopping | Flipkart | 47,840 |
| Sports | Billiards City | 32,280 |
| Video players & editors | MX Player | 20,781 |
| Weather | Weather & Clock widget | 27,324 |

**Table 7  Results of learning models for numeric rating prediction using TF features.**

| Models | Accuracy | Precision | Recall | F1-Score |
|---|---|---|---|---|
| XGB | 74% | 71% | 73% | 72% |
| RF | 71% | 73% | 77% | 75% |
| GBM | 70% | 68% | 71% | 70% |
| AB | 73% | 75% | 77% | 76% |
| ET | 75% | 79% | 82% | 81% |

**Table 8  Results of learning models for numeric rating prediction using TF/IDF features.**

| Models | Accuracy | Precision | Recall | F1-Score |
|---|---|---|---|---|
| XGB | 77% | 81% | 81% | 81% |
| RF | 81% | 83% | 86% | 84% |
| GBM | 82% | 86% | 89% | 88% |
| AB | 82% | 84% | 84% | 84% |
| ET | 85% | 89% | 91% | 90% |

The results of machine learning models using TF-IDF features are presented in Table 8. The results demonstrate that the performance of these models improves when utilizing TF-IDF features. TF-IDF, which identifies weighted features, enhances the feature vector used for training the machine learning models, whereas TF alone provides a basic feature vector. Once again, the ET model exhibits superior performance compared to other models in terms of accuracy, precision, recall, and F1 score when using TF-IDF features, achieving an accuracy rate of 85%. GBM and AB models closely follow with an accuracy of 82%. However, XGB performs poorly when employed with TF-IDF features.

**Table 9   Performance evaluation of transfer learning models.**

| Models | Accuracy | Precision | Recall | F1-Score |
|---|---|---|---|---|
| BERT | 94% | 93% | 95% | 94% |
| RoBERTa | 92% | 90% | 94% | 92% |
| XLNet | 93% | 92% | 93% | 92% |
| ELMo | 96% | 94% | 98% | 96% |

**Table 10   A few examples of biased application rating.**

| User review | User app rating |
|---|---|
| 'This app is a pure waste of time, ads consume a lot of time, ads are too much irritating' | 4 |
| 'A lot of nudity in the ads. Especially the ad in which the woman shaking her a$$' | 5 |
| 'I am uninstalling this game just because of a prostitution ad coming up on the screen .' | 5 |
| 'This app consumes a lot of time to load' | 4 |
| 'Plsss fix up the issues, app is taking too much time to load' | 4 |

In the third set of experiments, transfer learning models are employed. Four transfer learning models are utilized, and their performance is evaluated based on accuracy, precision, recall, and F1 score. The outcomes of these transfer learning models are given in Table 9. The results indicate that the transfer learning model ELMo outperforms all the other learning models used in the study, achieving an impressive accuracy of 96%. It is followed by the BERT which achieved an accuracy of 94%. Other learning models, XLNet and RoBERTa achieved accuracy of 93% and 92%, respectively.

## Methodology adopted to evaluate prediction performance

Since higher ratings tend to attract more new users, user ratings on the Android Play store can be prejudiced or even extravagant. Table 10 provides examples that demonstrate a significant difference between reviews and ratings. However, a systematic evaluation of this phenomenon is required to gain a more comprehensive understanding.

This study introduces an algorithm that utilizes TextBlob to determine the points of divergence between user reviews and an app's rating. Figure 9 presents the flowchart illustrating the methodology adopted in this study. The implementation of this technique is done using Python, resulting in the creation of a named tuple called Sentiment (polarity, subjectivity) obtained from the sentiment property of TextBlob (*Loria, 2018*). The subjectivity score is a floating-point number ranging from 0.0 to 1.0, where 1.0 indicates high subjectivity and 0.0 represents strong objectivity. The polarity score ranging from 1.0 to 1.0, is also a floating-point number. For example

from textblob import TextBlob

*tbl = TextBlob("He is honest. i love him")*

*tbl.sentiment()*

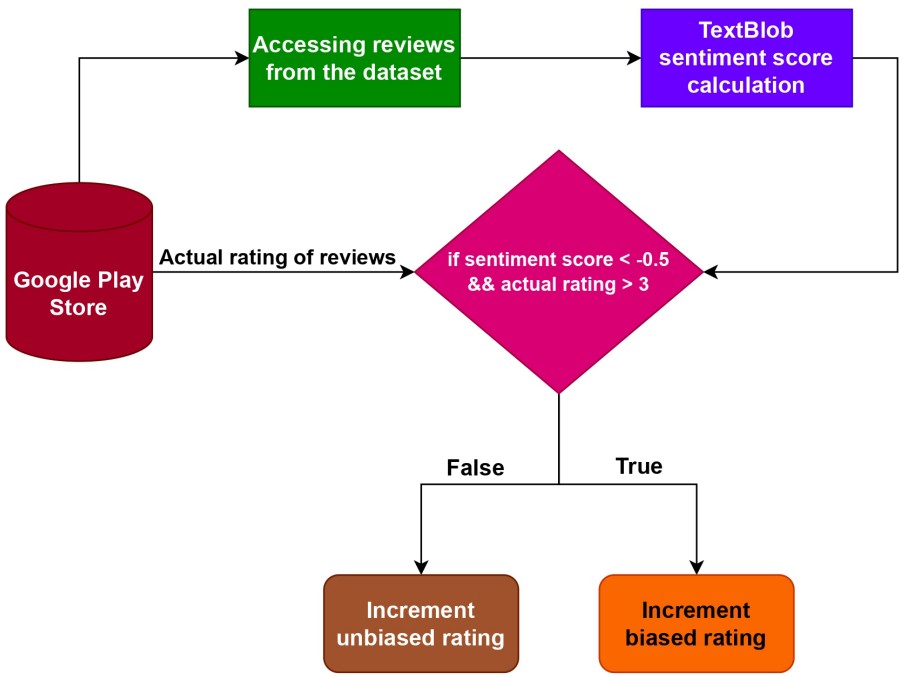

**Figure 9** **Proposed methodology to predict the biased numeric rating of applications.**

The output of the above code is a sentiment polarity value of 0.9 and a subjectivity value of 0.7)

To detect possible biases, polarity calculations are performed on every review and then compared to the corresponding rating in the dataset. If the polarity of a review is found to be less than 0.5 and the associated rating is over 3, we consider the rating by the user to be biased. Conversely, if the polarity of the review is equal to or greater than 0.5, or the rating is less than or equal to 3, we consider the rating to be impartial. This approach allowed us to identify potential biases based on the relationship between the review's polarity and the attached rating.

The experimental results of the proposed approach are presented in Table 11. Considering high ratings in influencing new users, this study specifically investigates the discrepancies in 3-star and above ratings and their equivalent evaluations. Among the analyzed 502,658 user ratings, the research identifies that 124,238 rating counts exhibited biases. This implies that approximately 24.7162% of the overall ratings are biased, within the selected app categories while the remaining 75.2838% represent unbiased reviews.

### Google apps reviews numeric rating prediction using transfer learning models

As we can observe from Tables 7, 8 and 9, the best performing models are ElMo, BERT, XLNet, and RoBERTa. So, for phase 2 this research will make use of these transfer learning models for numeric rating prediction by analyzing user reviews. These predicted numeric ratings are compared to the overall ratings (aggregate ratings) of the respective applications

**Table 11  TextBlob calculated biased rating results.**

| Application numeric rating | Count of numeric ratings | Biased rating count | Actual rating count |
|---|---|---|---|
| 5 | 336,781 | 89,034 | 247,747 |
| 4 | 68,419 | 20,479 | 47,940 |
| 3 | 32,238 | 14,725 | 17,513 |
| 2 | 15,605 | 0 | 15,605 |
| 1 | 49,608 | 0 | 49,608 |

**Table 12  Biased numeric detection using transfer learning models.** Bold phrases indicate classifier predictions that are closest to the actual ratings.

| Application name | Application reviews | XLNet | RoBERTa | BERT | ELMO | Aggregate rating |
|---|---|---|---|---|---|---|
| Beach Buggy Racing | 4,480 | 3.25 | 3.67 | 3.75 | **3.79** | 4.61 |
| B612 | 4,000 | 3.11 | 3.37 | 3.42 | **3.79** | 4.64 |
| Billiards City | 4,480 | 3.47 | 3.72 | 3.85 | **4.02** | 4.47 |
| Candy Crush Saga | 4,480 | 3.14 | 3.34 | 3.45 | **3.62** | 4.56 |
| Flipkart | 4,480 | 2.67 | 2.75 | 2.83 | **2.90** | 3.80 |
| Gun Shot | 3,000 | 3.15 | 3.24 | 3.37 | **3.40** | 3.69 |
| MX Player | 3,000 | 3.05 | 3.17 | 3.20 | **3.32** | 3.92 |
| Teen Pati Gold | 4,480 | 3.74 | 3.98 | 4.13 | **4.19** | 4.61 |
| Pharmapedia Pakistan | 4,133 | 3.38 | 3.47 | 3.57 | **3.76** | 4.72 |
| PhonePe | 4,000 | 3.17 | 3.32 | 3.42 | **3.45** | 3.98 |
| Seven | 4,424 | 3.25 | 3.41 | 3.52 | **3.72** | 4.43 |
| Temple Run 2 | 3,000 | 4.04 | 4.09 | 4.14 | **4.19** | 4.45 |
| UC-Browser | 3,000 | 3.32 | 3.34 | 3.35 | **3.46** | 4.20 |
| Weather & Clock Widget | 4,480 | 3.27 | 3.35 | 3.48 | **3.59** | 4.33 |

on the Google Play Store. This comparison is done to identify any inconsistencies between the ratings and the user reviews. Table 12 presents the numeric ratings prediction by the transfer learning models selected for the experiment.

Bold phrases indicate classifier predictions that are closest to the actual ratings. The bold figures in Table 12 represent the terms that are expected to have the highest ratings. It is noteworthy that the ELMo classifier predicts the highest scores for 14 out of the 14 categories. Additionally, ELMo's projected ratings are relatively closer to the aggregate ratings compared to the other classifiers.

The Google applications with the lowest expected ratings are XLNet and RoBERTa. Both these models are computationally complex and also require a large amount of data for training. XLNet also faces scalability issues with lesser data as in the current case. While RoBERTa requires fine-tuning of hyperparameters according to the nature of the data. These are the major concerns that cause the lower rating prediction of Google app data from these two models.

The results of the ELMo learning model reveal the differences between the user-specified numeric rating and the accompanying reviews. It is observed that the numeric rating

provided by users is approximately 25% higher in accuracy compared to the predictions generated by ELMo.

## CONCLUSION

This study explores the significance of user reviews and ratings in assessing app quality on platforms like Google Play. User reviews offer qualitative insights, while ratings provide a quantitative measure. However, the growing volume of review-based data necessitates the use of predictions to extract valuable information. This study focuses on predicting app ratings based on user evaluations in the Google Play store, using transfer learning models. To evaluate the effectiveness of these models, a comprehensive analysis is conducted using a dataset of scraped reviews from Google Play. We used machine learning classifiers with TF and TF/IDF features for prediction, and TextBlob analysis to identify reviews with discrepancies between the user-assigned ratings and the review text, creating a valuable ground truth for our evaluation. The result findings revealed that approximately 25.64% of user-defined app ratings are considered unreliable based on TextBlob analysis. Notably, transfer learning classifiers outperformed traditional machine learning classifiers significantly, demonstrating their ability to handle nonlinearity, colinearity, and data noise. This suggests that transfer learning can lead to more accurate predictions of app ratings based on user reviews.

Furthermore, this study shed light on the inconsistency between user reviews and ratings, with ratings often being higher than what the reviews would suggest. As a potential avenue for future research, we propose exploring ensemble approaches that combine machine and deep learning techniques. Deep machine ensemble learning has the potential to capture complex patterns and relationships in textual data, promising even greater accuracy in rating predictions. This study contributes to the field of sentiment analysis and app rating prediction by highlighting the efficacy of transfer learning models and the potential for future research in employing deep machine ensemble techniques. These insights can aid in improving the assessment of app quality and user satisfaction in the ever-evolving landscape of mobile applications.

## ACKNOWLEDGEMENTS

The authors are deeply grateful to all those who contributed to this article and those who played a big role in the success of this article. We would like to thank the University of Hafr Al Batin and Prince Satam bin Abdulaziz for their invaluable support for providing research systems and tools to design this research work.

### Funding

This article was funded by Princess Nourah bint Abdulrahman University Researchers Supporting Project number (PNURSP2024R349), Princess Nourah bint Abdulrahman

University, Riyadh, Saudi Arabia. The funders had no role in study design, data collection and analysis, decision to publish, or preparation of the manuscript.

### Grant Disclosures

The following grant information was disclosed by the authors:

Princess Nourah bint Abdulrahman University, Riyadh, Saudi Arabia: PNURSP2024R349.

### Competing Interests

Imran Ashraf is an Academic Editor for PeerJ.

### Author Contributions

- Turki Aljrees conceived and designed the experiments, analyzed the data, authored or reviewed drafts of the article, and approved the final draft.
- Muhammad Umer conceived and designed the experiments, performed the experiments, performed the computation work, authored or reviewed drafts of the article, and approved the final draft.
- Oumaima Saidani performed the experiments, analyzed the data, prepared figures and/or tables, authored or reviewed drafts of the article, and approved the final draft.
- Latifah Almuqren performed the experiments, analyzed the data, prepared figures and/or tables, and approved the final draft.
- Abid Ishaq conceived and designed the experiments, performed the experiments, authored or reviewed drafts of the article, and approved the final draft.
- Shtwai Alsubai conceived and designed the experiments, performed the computation work, prepared figures and/or tables, and approved the final draft.
- Ala' Abdulmajid Eshmawi conceived and designed the experiments, performed the computation work, prepared figures and/or tables, and approved the final draft.
- Imran Ashraf analyzed the data, performed the computation work, prepared figures and/or tables, authored or reviewed drafts of the article, and approved the final draft.

### Data Availability

The data are available at GitHub and Zenodo:

- https://github.com/MUmerSabir/GoogleApps_PeerJCS

- MUmerSabir. (2023). MUmerSabir/GoogleApps_PeerJCS: DOI Request zenodo (main). Zenodo. https://doi.org/10.5281/zenodo.8132355

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
