# Peer review of "Contradiction in text review and apps rating: prediction using textual features and transfer learning"

_PeerJ Computer Science, doi:10.7717/peerj-cs.1722_

## Round 0.1 · original submission · Major Revisions

Based on the comments obtained by the reviewers, it has been decided to request major changes for the acceptance of the manuscript. Please pay attention to all comments made by the reviewers.

Reviewer 1 has suggested that you cite specific references. You are welcome to add it/them if you believe they are relevant. However, you are not required to include these citations, and if you do not include them, this will not influence my decision.

**Language Note:** The review process has identified that the English language must be improved. PeerJ can provide language editing services - please contact us at copyediting@peerj.com for pricing (be sure to provide your manuscript number and title). Alternatively, you should make your own arrangements to improve the language quality and provide details in your response letter. – PeerJ Staff

Reviewer 1 ·

Basic reporting

The manuscript entitled " Contradiction in text review and apps rating: prediction using textual features and transfer learning” describe a model for predicting real digital ratings of user reviews. Their work is very meaningful and valuable in the field. However, some issues still should be addressed.
1. The abstract should identify the research gap and propose the academic contribution of the paper, showing numerical results.
2. The related works section is too limited. Improving this section with the latest works may increase the quality of the paper. I could recommend you to read and integrated the following articles. https://doi.org/10.3390/systems11080390, https://doi.org/10.1057/s41599-023-01816-6, https://doi.org/10.3390/electronics11193022, https://doi.org/10.1016/j.cie.2022.107939, https://doi.org/10.1016/j.tmp.2022.101063, 10.1109/ACCESS.2023.3246388.
3.It is strongly suggested that the references should be renewed, most of the references should be published within 5 years, as the domain develops rapidly.

Experimental design

It is an interesting work, and the manuscript is well organized. The experimental design is reasonable and well-described. However, it is suggested that the authors add a diagram (or set of diagrams) to illustrate the model structure being used in this manuscript.

Validity of the findings

No comment.

Additional comments

The conclusion of this paper needs to be optimized. The authors may give the details of their paper's novelty with short descriptions. It is suggested that the author add some comparisons with previous work, advantages and disadvantages of the author's method, and prospects for future research directions.

Reviewer 2 ·

Basic reporting

The paper first provides background on the evolution of apps on the Google Play Store for the recent decade and shows that biased or fraudulent reviews can lure users to download some apps based on their star ratings. Then, the paper provides a deeper background on the models used for predicting numeric ratings. Furthermore, the proposed approach for the applications rating evaluation shows the experiment and results. Also, the paper offers a clear motivation and clear definitions of all terms and theorems.

Experimental design

In this paper, the authors categorize ratings of different kinds of apps (such as sports, action, arcade, video players & editors, etc.) and analyze the framework of data scrapping extensively. The categories made by this paper are easy to follow and have a clear logical structure. The way of describing different existing solutions lets people know their special properties and limitations.

Validity of the findings

As a result of this work, one major novelty is that the authors emphasized utilizing transfer learning models. The article compares and critically examines the investigated solutions by examining their methods to predict app ratings based on user evaluations in the Google Play Store. At the end of the article, the authors identify possible future research directions for improving their learning models. The conclusion part is appropriately stated and connects to the original question investigated.

Additional comments

The introduction of "XLNet" needs more detail. I suggest that you improve the description and add more references about how "XLNet" addresses the limitations of autoencoder (AE) models.

Please add more references to some statements made in the article, such as the numerical ratings’ legitimacy, the features of hand-crafted TF/IDF, and the pre-trained word vector representation of bidirectional encoder representations from Transformers (BERT).

Please explain more about how to measure the accuracy, precision, recall, and F1-score in Table 8, such as the difference between accuracy and precision.

---

## Round 0.2 · Minor Revisions

Dear authors:

All the Reviewer 1's suggestion have been addressed.

I detected irregularities related to the literature proposed in last review, so, please, remove these references:

[19] Liu, X., Shi, T., Zhou, G., Liu, M., Yin, Z., Yin, L., and Zheng, W. (2023a). Emotion classification for short texts: an improved multi-label method. Humanities and Social Sciences Communications, 10(1):1–9.

[20] Liu, X., Zhou, G., Kong, M., Yin, Z., Li, X., Yin, L., and Zheng, W. (2023b). Developing multi labelled corpus of twitter short texts: a semi-automatic method. Systems, 11(8):390.

[28] Qin, X., Liu, Z., Liu, Y., Liu, S., Yang, B., Yin, L., Liu, M., and Zheng, W. (2022). User ocean personality model construction method using a bp neural network. Electronics, 11(19):3022.

In addition to this change, please, pay attention to the new minor suggestions proposed by reviewer 2.

Reviewer 2 ·

Basic reporting

No further comments.

Experimental design

No further comments.

Validity of the findings

No further comments.

Additional comments

In the first round, I suggested the paper add more references to several critical statements. I gave examples such as the numerical ratings’ legitimacy, hand-crafted TF/IDF features, etc. It's good to see that the revised version added some references, but there's still room to improve the references in the introduction section:

(line 48) popular opinion mining techniques, examples of using computational linguistics, text analysis, and natural language processing;
(line 55) Users’ and app developers’ participation broadens with issue reports (such as the existing work "DroidPerf: Profiling Memory Objects on Android Devices" [MobiCom'23] shows a lot of reported performance issues from user experiences);
(line 64) App examples of biased reviews lure users to download.

---

## Round 0.3 · accepted · Accept

The authors have addressed all of the reviewer's comments.
I have checked the revision myself and it is good.

The manuscript is ready for publication.
Congrats to the authors.